# Exploration of microbial communities contributing to effective methane production from scum under anaerobic digestion

Riku Sakurai[1], Shuhei Takizawa[1,2¤], Yasuhiro Fukuda[1], Chika Tada[1] *

1 Laboratory of Sustainable Animal Environment, Graduate School of Agricultural Science, Tohoku University, Osaki, Miyagi, Japan, 2 Research Fellow of Japan Society for the Promotion of Science, Japan Society for the Promotion of Science, Chiyoda-ku, Tokyo, Japan

¤ Current address: Institute of Livestock and Grassland Science, National Agriculture and Food Research Organization, Tsukuba, Ibaraki, Japan

* chika.tada.e1@tohoku.ac.jp

**Data Availability Statement:** Raw sequence data will be available only after acceptance of the manuscript for publication.

## Abstract

Scum is formed by the adsorption of long-chain fatty acids (LCFAs) onto biomass surface in anaerobic digestion of oily substrates. Since scum is a recalcitrant substrate to be digested, it is disposed via landfilling or incineration, which results in biomass washout and a decrease in methane yield. The microbes contributing to scum degradation are unclear. This study aimed to investigate the cardinal microorganisms in anaerobic scum digestion. We pre-incubated a sludge with scum to enrich scum-degrading microbes. Using this sludge, a 1.3-times higher methane conversion rate (73%) and a faster LCFA degradation compared with control sludge were attained. Then, we analyzed the cardinal scum-degrading microbes in this pre-incubated sludge by changing the initial scum-loading rates. Increased 16S rRNA copy numbers for the syntrophic fatty-acid degrader *Syntrophomonas* and hydrogenotrophic methanogens were observed in scum high-loaded samples. 16S rRNA amplicon sequencing indicated that *Syntrophomonas* was the most abundant genus in all the samples. The amino-acid degrader *Aminobacterium* and hydrolytic genera such as *Defluviitoga* and *Sporanaerobacter* became more dominant as the scum-loading rate increased. Moreover, phylogenic analysis on *Syntrophomonas* revealed that *Syntrophomonas palmitatica*, which is capable of degrading LCFAs, related species became more dominant as the scum-loading rate increased. These results indicate that a variety of microorganisms that degrade LCFAs, proteins, and sugars are involved in effective scum degradation.

## Introduction

Anaerobic digestion can produce biomethane as an energy resource from a variety of organic wastes. Wastewater from food processing, edible oil producers, and slaughterhouses contains high concentrations of lipids [1]. Lipids have higher theoretical methane yields (1.01 m³ kg⁻¹ VS) than that of carbohydrates (e.g., 0.37 m³ kg⁻¹ VS for glucose) and proteins (0.74 m³ kg⁻¹ VS) [2]. However, there are several challenges in the anaerobic digestion of lipidic waste. Scum

**Funding:** This research was funded by the Project of Integrated Compost Science in Tohoku University and Bio Energy Corporation in Japan, Project of Integrated Compost Science (PICS), and the Pacific Institute for Climate Solutions awarded to C.A.

**Competing interests:** The authors have declared that no competing interests exist.

formation is one of the most severe problems. Scum is formed by the adsorption of floating substrates such as long-chain fatty acids (LCFAs) to the surface of biomass [1, 3]. Scum is a recalcitrant substrate to be digested and inhibits the liquidity in the reactor. Accumulated scum must be periodically removed and disposed via landfills or incineration, which leads to the washout of organic waste. Scum formation and following biomass washout disturb biomethane production by reducing substrate utilization rates. The main cause of scum formation is LCFAs, which are produced by the hydrolysis of lipids. LCFAs are degraded to volatile fatty acids (VFAs). This is the rate-limiting step of the anaerobic digestion process, as the reaction proceeds only under very low $H_2$ pressure [4]. Moreover, LCFAs have been considered to adsorb onto the microbial surface and inhibit the activities of anaerobic microorganisms [5]. Previous reports showed that 0.5 mmol $L^{-1}$ oleic acid (C18:1) or 2–4 mmol $L^{-1}$ palmitic acid (C16:0) reduced methane generation by more than 50% [6]. Despite these troublesome characteristics of LCFAs, it has been reported that scum formation leading to biomass washout contributed more to the failure of anaerobic digestion than the toxicity of LCFAs themselves [1]. Biomass washout severely impairs the high methane convertibility of lipids.

To reuse the collected scum as energy resources, several studies have employed scum as a co-digestion substrate with sewage sludge [7], thickened activated sludge, and primary sludge [3, 8]. The development of efficient scum digestion methods enables reduction of the cost of disposal and increase in biomethane yield. If anaerobic scum digestion can be promoted, the application range of anaerobic digestion can be expanded to include more oily substrates. However, the key microbes that play an important role in anaerobic scum digestion are unclear.

It has been reported that pre-incubation with lipidic substrates is an effective strategy for improving methane yield. A previous study reported that pre-incubation with soybean oil-based wastewater increased the relative abundance of Synergistales, which was highly correlated with the methane production rate [9]. Ziels et al., (2016) showed that acclimation with waste restaurant oil increased the relative abundance of the genus *Syntrophomo*nas, which has the capability of degrading various fatty acids [10]. The substrate-loading rate also affects the microbial community. Ziels et al., (2016) showed that an increase in the loading rates of lipidic substrates positively affected the abundance of *Syntrophomonas* 16S rRNA genes [10]. It is assumed that a higher loading rate of the substrate increases the core microbial population that plays an important role in substrate degradation.

Our microbiological findings on anaerobic scum digestion will enable new approaches to solve the problems caused by scum. In this study, we aimed to reveal the microbes contributing to effective scum degradation. We pre-incubated a collected sludge with scum to enrich scum-degrading microbes. Additionally, to investigate the core scum-degrading microbes in this pre-incubated sludge, we evaluated the changes in microbial communities according to various loading rates.

## Materials and methods

### Preparation of sludges

Seed sludge was collected from a mesophilic food waste treatment plant at Tohoku University (Osaki, Japan). Before incubation with scum, this sludge was incubated with olive oil (1 mL $L^{-1}$) at 35°C in a 2-L screwcap bottle (1,800 L working volume). The consumption of the substrate was confirmed by the cessation of biogas generation [11]. When biogas production ceased, 2/3 of the culture was exchanged with fresh medium and seed sludge (1:1 v/v). The medium consisted of 0.14 g $L^{-1}$ of $KH_2PO_4$, 0.54 g $L^{-1}$ of $NH_4Cl$, 0.20 g $L^{-1}$ of $MgCl_2·6H_2O$, 0.15 g $L^{-1}$ of $CaCl_2·2H_2O$, 2.5 g $L^{-1}$ of $NaHCO_3$, and 0.20 g $L^{-1}$ of yeast extract (Difco, Detroit,

MI, USA). Trace element and vitamin solutions were also used; details are provided in NBRC No. 398 (NITE Biological Resource Center, Chiba, Japan). After approximately 2 years of incubation, the olive oil was replaced with scum, which was collected from the oil separation tank before treatment of the food wastewater. The chemical oxygen demand (COD) of the scum was 280 g kg$^{-1}$. Granular activated carbon was then added to promote methane production [12], and the sludge was incubated at 35˚C for approximately 6 months (Sludge I). Another sludge was collected from a mesophilic biogas plant at Tohoku University (Osaki, Japan) and incubated with 2 g L$^{-1}$ glucose at 35˚C until use to avoid starvation. After cessation of biogas production was observed, this sludge was used in Experiment 1 (Sludge II).

## Experiment 1: Assessment of the digestibility of scum

The scum used for the batch experiments was collected from a mesophilic food waste treatment plant (Tokyo, Japan) and stored at 4˚C. The characteristics of scum and sludge are shown in Table 1. A total of 40 mL of Sludge I or Sludge II, 3.0 g of granular activated carbon, and 40 mL of NBRC No. 398 medium mentioned above were placed into a 100-mL vial. Scum was then added to adjust the concentration of 8.6 g COD L$^{-1}$. The vials were purged with nitrogen gas to remove oxygen. The batch experiments were conducted in sextuplicate until day 8, and three of these replicates were opened on day 8 for DNA extraction and LCFA analysis. Then the batch experiments were continued using the remaining triplicates.

## Experiment 2: Exploration of core scum-digesting microbes

Batch experiments were conducted in parallel under the same conditions as those employed in Experiment 1. A total of 40 mL of Sludge I, 3.0 g of granular activated carbon, and 40 mL of NBRC No. 398 medium mentioned above were placed into a 100-mL vial. The initial scum loading concentrations were as follows: 8.6 g COD L$^{-1}$, 12.6 g COD L$^{-1}$, and 17.3 g COD L$^{-1}$. The batch experiments were conducted in sextuplicate until day 8, and three of these replicates were opened on day 8 for DNA extraction and LCFA analysis. Then, the batch experiments were continued using the remaining triplicates. A schematic representation of this study is given in Fig 1.

## Chemical analysis

COD was evaluated by the colorimetric method using HACH Digestion Solution for COD with a range of 0–1500 ppm (HACH, Loveland, USA). VFA concentrations were measured using high-performance liquid chromatography (HPLC) (Jasco, Tokyo, Japan), with an ion-exchange column (RSpak KC-811; Shodex, Tokyo, Japan) and an ultraviolet detector (870-UV; Jasco). The column temperature was 60˚C, the eluent consisted of 90% (v/v) 3 mM perchloric acid and 10% (v/v) acetonitrile, and the flow rate was 1.0 mL min$^{-1}$ [13]. Ester-type and free-type LCFA (oleate, stearate, and palmitate) concentrations were measured by HPLC using a labeling reagent obtained from YMC Co. (Kyoto, Japan) according to the

**Table 1. Characteristics of sludges and scum.**

|  | Sludge I (g L$^{-1}$) | Sludge II (g L$^{-1}$) | Scum (g kg$^{-1}$) |
|---|---|---|---|
| Total solids (TS) | 2.0 ± 0.0 | 3.1 ± 0.0 | 115.0 ± 3.0 |
| Volatile solids (VS) | 1.3 ± 0.0 | 1.8 ± 0.1 | 100.0 ± 2.5 |
| Total COD | 0.187 ± 0.0 | 0.442 ± 0.0 | 223 ± 0.0 |

Values represent the standard deviation of the mean (n = 3 for TS and VS, n = 2 for COD).

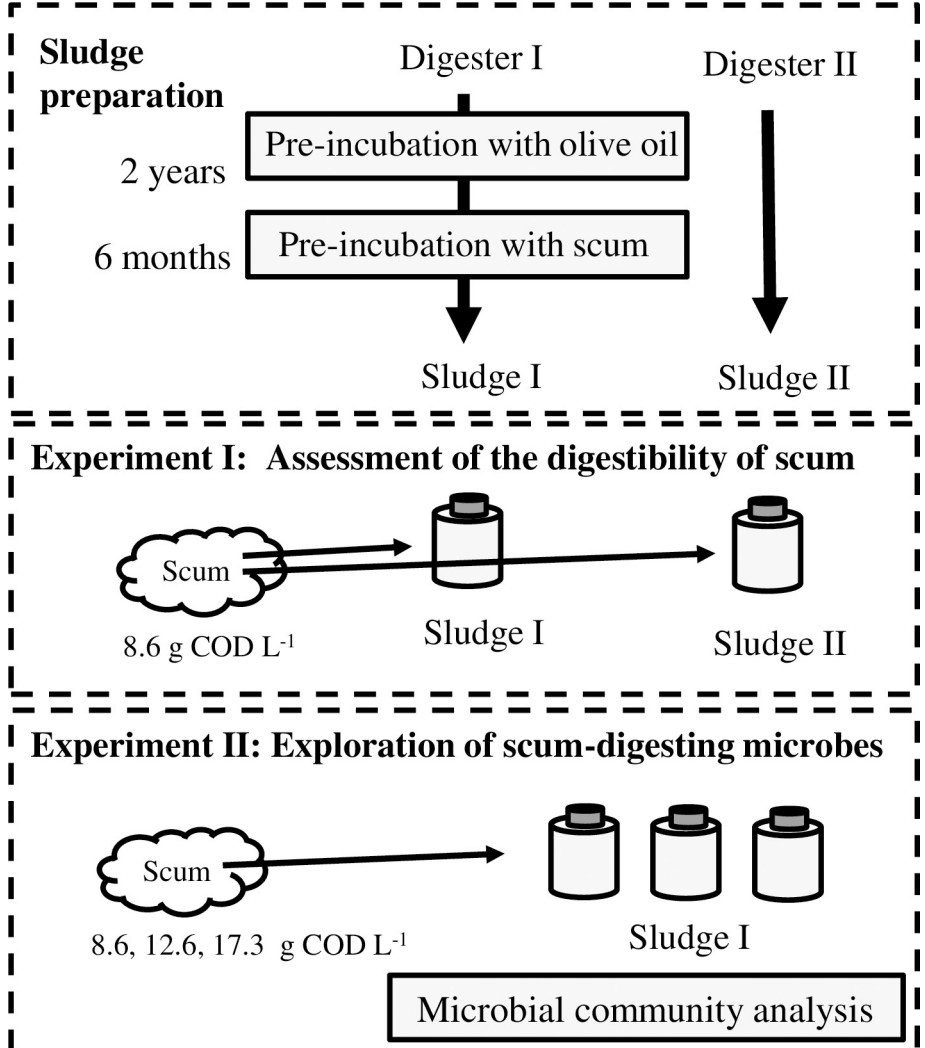

**Fig 1. Schematic workflow for the exploration of scum-digesting microbes.**

manufacturer's protocol. Methyl oleate, methyl stearate, and methyl palmitate were used as the standard mixtures, and the extraction efficiency was confirmed ($R_2 > 0.99$). The TS and VS of the sludge scum were measured by heating the samples at 100°C and 600°C overnight, respectively. The biogas volume was measured using a syringe, and the methane concentration was determined by gas chromatography (GC-8A; Shimadzu, Kyoto, Japan). The injection and detection temperatures were set to 100°C and 120°C, respectively. A packed column (Shincarbon-ST; Restek, Bellefonte, PA, USA) was used, and a unit equipped with a thermal conductivity detector was connected to an integrator (C-R8A; Shimadzu). Argon was used as the carrier gas. A modified Gompertz equation was applied to analyze methanogenesis performance quantitatively [14].

$$M(t) = Mmax \times \exp\{-\exp[Rm \times e/Mmax \times (\lambda - t) + 1]\} \qquad (1)$$

M(t) is the cumulative methane production at t (h), Mmax is the total amount of methane produced in time t, Rm is the maximum methane production rate, $\lambda$ is the lag phase, t is the incubation time (d), and e is the exponential constant.

## Real-time polymerase chain reaction (PCR)

DNA extraction was conducted using Power Soil DNA Isolation Kit (MO BIO Laboratory, Carlsbad, CA, USA) according to the manufacturer's instructions. Real-time PCR was conducted using primer sets targeting two orders of hydrogenotrophic methanogens (Methanobacteriales and Methanomicrobiales), two families of acetoclastic methanogens (Methanosarcinaceae and Methanosaetaceae) [15], and genus *Syntrophomonas* (S1 Table) [16]. Probe qPCR Mix (TaKaRa Bio) was used for PCR. The samples were assayed in triplicate in a 25-μL reaction mixture, which consisted of 12.5 μL of Probe qPCR Mix, 0.25 μL of 20 μmol $L^{-1}$ forward primer, reverse primer, and TaqMan probe, 2 μL of template, and 9.75 μL of sterile water. The PCR conditions for Methanosaetaceae and Methanosarcinaceae were as follows: 95˚C for 30 s, followed by 45 cycles at 95˚C for 10 s, and 60˚C for 30 s. The PCR conditions for Methanomicrobiales were as follows: 95˚C for 30 s, followed by 40 cycles at 95˚C for 10 s, and 63˚C for 30 s. The PCR conditions for Methanobacteriales were as follows: 95˚C for 10 min, followed by 45 cycles at 95˚C for 10 s and 60˚C for 30 s. The PCR conditions for *Syntrophomonas* were as follows: 94˚C for 10 min, followed by 35 cycles at 94˚C 10 s and 51˚C for 5 s and 72˚C 10 s. All real-time PCR amplification and detection were performed using a Thermal Cycler Dice real-time system (TaKaRa Bio).

## Bacterial community analysis

DNA was extracted as described above. As for the extracted DNA on day 8, the bacterial community was analyzed using 16S rRNA gene amplicon sequencing using MiSeq technology (Illumina, San Diego, USA). DNA concentrations were analyzed using Synergy H1 (Bio Tek, Winooski, USA) and QuantiFluor dsDNA System (Promega, Madison, WI, USA). The 16S rRNA gene fragments were amplified via two-step tailed PCR using bacterial V3-V4 primers 341 F (5′–CCTACGGGNGGCWGCAG–3′) and 805 R (5′–GACTACHVGGGTATCTAATCC–3′) [17]. The reaction mixture consisted of 1 μL of Ex buffer, 0.8 μL of dNTPs, 0.5 μL of 10 μmol $L^{-1}$ forward primer and reverse primer, 2 μL of template DNA, 0.1 μL of Ex Taq HS (TaKaRa Bio, Shiga, Japan), and 5.1 μL of sterile water. The first PCR conditions were as follows: 94˚C for 2 min, followed by 30 cycles at 94˚C for 30 s, 55˚C for 30 s, 72˚C for 30 s, and 72˚C for 5 min. The second PCR conditions were as follows: 94˚C for 2 min, followed by 10 cycles at 94˚C for 30 s, 60˚C for 30 s, 72˚C for 30 s, and 72˚C for 5 min. The PCR products of the first and second PCRs were purified using AMPureXP (Beckman Coulter, Brea, CA, USA). Library quality was confirmed using a fragment analyzer and dsDNA 915 Reagent Kit (Advanced Analytical Technologies, Santa Clara, CA, USA). Subsequently, the products were pooled and sequenced using 2×300 bp Illumina MiSeq. The key tags and primers were trimmed, and sequences with quality scores less than 20 were removed using Pipeline Initial Process in Ribosomal Database Project (RDP) [18]. Sequences with lengths shorter than 150 or longer than 500 were excluded. Chimeric reads were excluded using RDP's FunGene [19]. The non-chimeric reads were used for further analysis using the RDP pipeline for microbial classification at a confidence cut-off of 80 [20]. The raw sequence data was submitted to the DDBJ Sequence Read Achieve as BioProject PRJDB12309.

Principal coordinate analysis (PCoA) was conducted using R package vegan version 2.5–6 based on Bray-Curtis dissimilarity (R version 3.6.1) [21].

## Phylogenetic analysis

The reads assigned to the *Syntrophomonas* genus were clustered into operational taxonomic units (OTUs) at 97% similarity using Vsearch [22]. The representative sequences and reference species in GenBank were aligned using the CLUSTAL X program (version 2.1) [23]. Gaps and

indistinguishable nucleotides were deleted using Jalview (version 2.9.0b2) [24]. Maximum likelihood (ML) analyses were performed using PhyML [25]. For the ML analysis, the best substitution model and optional parameters were evaluated using Kakusan4 (version 4.0) [26], and the Tamura-Nei 1993 model (TN93) was applied as the best setting [27]. The constructed ML tree was visualized using Figtree (version 1.4.4) [28].

## Results and discussion

### Experiment 1: Assessment of the digestibility of scum

The cumulative methane production of each sludge is shown in Fig 2A. Until day 7, the methane production of Sludge I was $30 \pm 1$ mL $g^{-1}$ VS. Then it increased to $121 \pm 1$ mL $g^{-1}$ VS on day 11. In contrast, the methane production of Sludge II was only $19 \pm 3$ mL $g^{-1}$ VS until day 11. The cumulative methane production on day 11 of Sludge I was 6-fold higher than that of Sludge II. Finally, $582 \pm 27$ and $457 \pm 5$ mL $g^{-1}$ VS methane were produced from Sludge I and Sludge II, respectively. Methane conversion rates of Sludge I and Sludge II calculated from COD were $73\% \pm 3\%$ and $57\% \pm 1\%$, respectively. Sludge I showed relatively high methane productivity from scum. The methane gasification process of scum was fitted with the modified Gompertz model in all samples (RMSE = $21 \pm 6$). The theoretical methane production of Sludge I and Sludge II calculated by the modified Gompertz model was $630 \pm 25$ and $470 \pm 6$ mL $g^{-1}$ VS, respectively. Fig 2B shows the transition of total VFA concentrations. On day 0, total VFA concentrations were $60 \pm 6$ and $87 \pm 4$ mg $L^{-1}$ in Sludge I and Sludge II, respectively. In Sludge I, VFA concentration then decreased gradually and became undetectable on day 22. In Sludge II, the VFA concentration gradually increased to $216 \pm 13$ mg $L^{-1}$ on day 15. Among them, acetate was the dominant VFA ($202 \pm 13$ mg $L^{-1}$). The concentration then decreased gradually and became undetectable on day 30. Considering the low methane production in Sludge II, it was inferred that acetate conversion to methane was inhibited until day 15. Methanogens are more susceptible to LCFA toxicity than acetogens [29]. In particular, acetoclastic

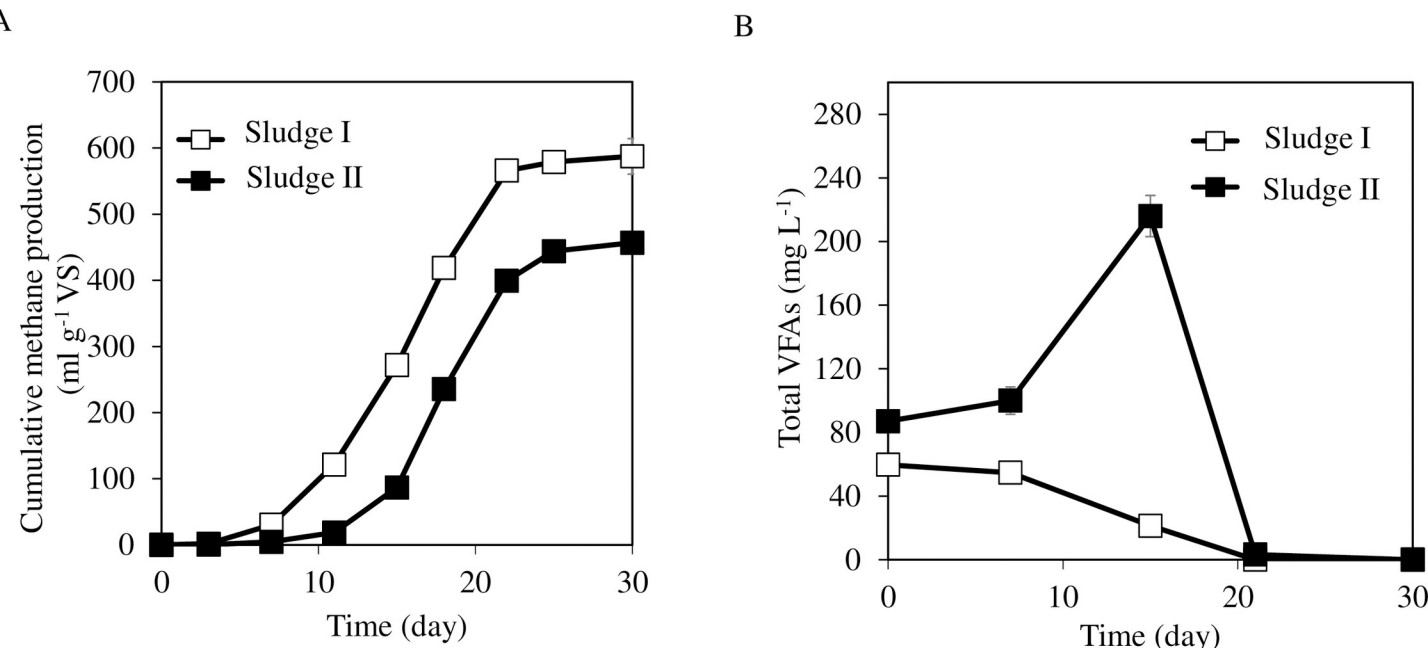

**Fig 2.** Cumulative methane production (A) and VFA concentration (B). Error bars represent the standard deviation of the mean (n = 3).

methanogens have been reported to be more sensitive to LCFAs than hydrogenotrophic methanogens [4, 30]. It was inferred that LCFAs degradation in Sludge I was faster than that in Sludge II.

The concentrations of total LCFAs were compared at two time points: day 8, when methane production was verified, and day 30, which was the end of our experimental fermentation (Fig 3). The total LCFA concentrations on day 8 in Sludge I and Sludge II were 1,142 ± 179 and 2,721 ± 198 mg L$^{-1}$, respectively. The total LCFA concentration in Sludge I was only 42% of that in Sludge II (T.TEST, p = 0.02). Finally, most LCFAs were degraded. On day 30, LCFA concentrations were 4 ± 0 and 4 ± 1 mg L$^{-1}$ in Sludge I and Sludge II, respectively. A previous study showed that palmitate inhibited anaerobic microorganisms with IC 50 values of more than 1,100 mg L$^{-1}$ [5]. The high concentration of palmitic acid might be one of the causes of the temporal VFA accumulation and the slower methane production in Sludge II (Fig 2). Though LCFAs and VFAs were almost completely digested in both sludges after the 30-day incubation, the methane conversion rate of Sludge I was 26% higher than that of Sludge II. Other substrates than fatty acids may have remained undegraded in Sludge II. The VS decomposition rate was higher in Sludge I (72.7% ± 0.9%) than in Sludge II (64.6% ± 0.4%) (T.TEST, p <0.05) (S1A Fig). The dissolved COD concentrations in Sludge I and Sludge II on day 30 were 68 ± 6 and 62 ± 3 mg L$^{-1}$, respectively, and there were no significant differences (T.TEST, p >0.05) (S1B Fig). These results suggest that non-dissolved, complex substances, such as proteins or polysaccharides, remained undigested in Sludge II. The rate of hydrolysis depends on several parameters such as pH, particle size, and the diffusion barrier between an enzyme and

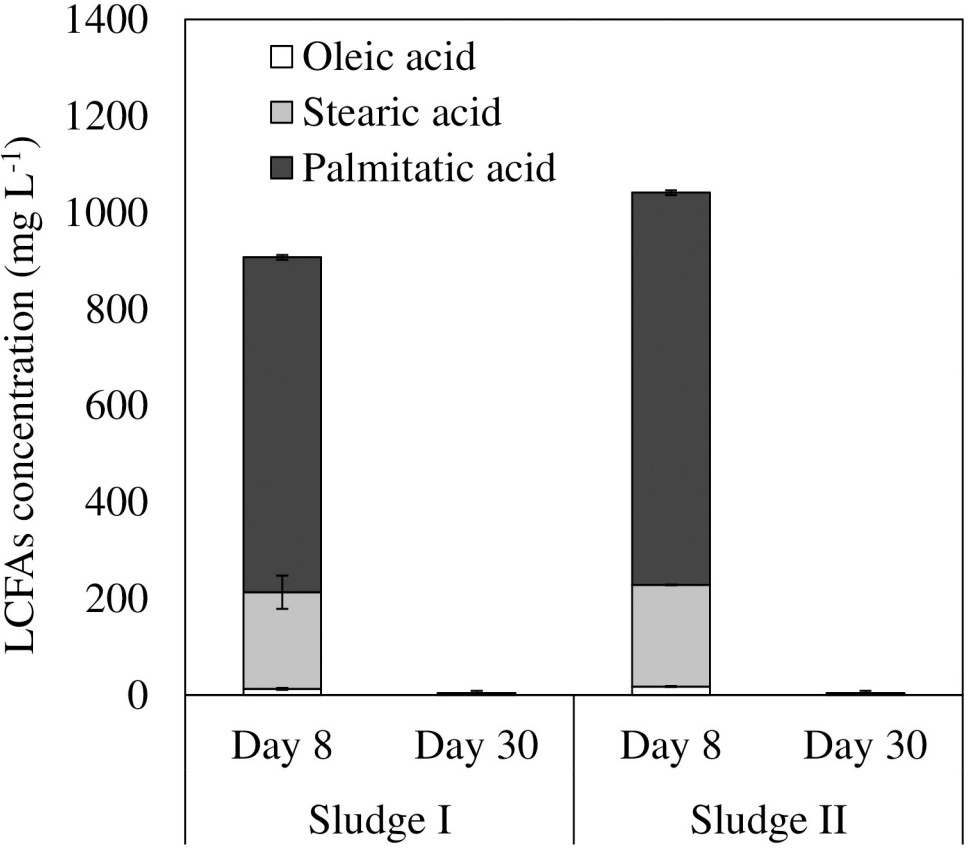

**Fig 3. The LCFAs concentration in each reactor.** Error bars represent the standard deviation of the mean (n = 3).

substrate [1]. LCFAs may have prevented hydrolytic enzymes from reacting with biomass. It was inferred that the degradation efficiency of LCFAs affected the degradation efficiency of other substrates.

Our results confirmed that Sludge I had a relatively high potential for methane gasification of scum. It was assumed that scum-degrading microorganisms were enriched in Sludge I. To reveal the core scum-digesting community, the differences in microbial communities were evaluated according to the scum loading concentration.

### Experiment 2: Exploration of core scum-digesting microbes

**The effect of scum loading concentration on methane productivity.** Cumulative methane production is shown in Fig 4A. During the 30-day incubation, $582 \pm 27$, $607 \pm 8$, and $563 \pm 22$ mL g$^{-1}$ VS methane were produced from 8.6, 12.6, and 17.3 g COD L$^{-1}$ scum-loaded vials, respectively. The methane gasification process of scum was well fitted with the modified Gompertz model in all samples (RMSE = $41 \pm 13$). The theoretical methane productions calculated by the modified Gompertz model were $643 \pm 37$, $667 \pm 7.3$, and $630 \pm 25$ in 8.6, 12.6, and 17.3 g COD L$^{-1}$ scum-loaded vials, respectively. The methane conversion rates in 8.6, 12.6, and 17.3 g COD L$^{-1}$ loaded vials were $73\% \pm 3\%$, $77\% \pm 1\%$, and $70\% \pm 3\%$, respectively. There were no significant differences between these values (TukeyHSD, p $>0.05$). The total VFAs gradually decreased in all the samples (Fig 4B). Temporal VFA accumulation was not observed. The total LCFA concentrations in 8.6, 12.6, and 17.3 g COD L$^{-1}$ loaded vials were $1,142 \pm 179$, $4,987 \pm 222$, and $6,869 \pm 585$ mg L$^{-1}$, respectively (S2 Fig). Finally, most LCFAs were degraded in all the samples. Total LCFAs on day 30 in 8.6, 12.6, and 17.3 g COD L$^{-1}$ loaded vials were $3.3 \pm 0.3$, $5.7 \pm 0.3$, and $19.3 \pm 8.0$ mg L$^{-1}$, respectively. Remarkably, the total

(A)                                                    (B)

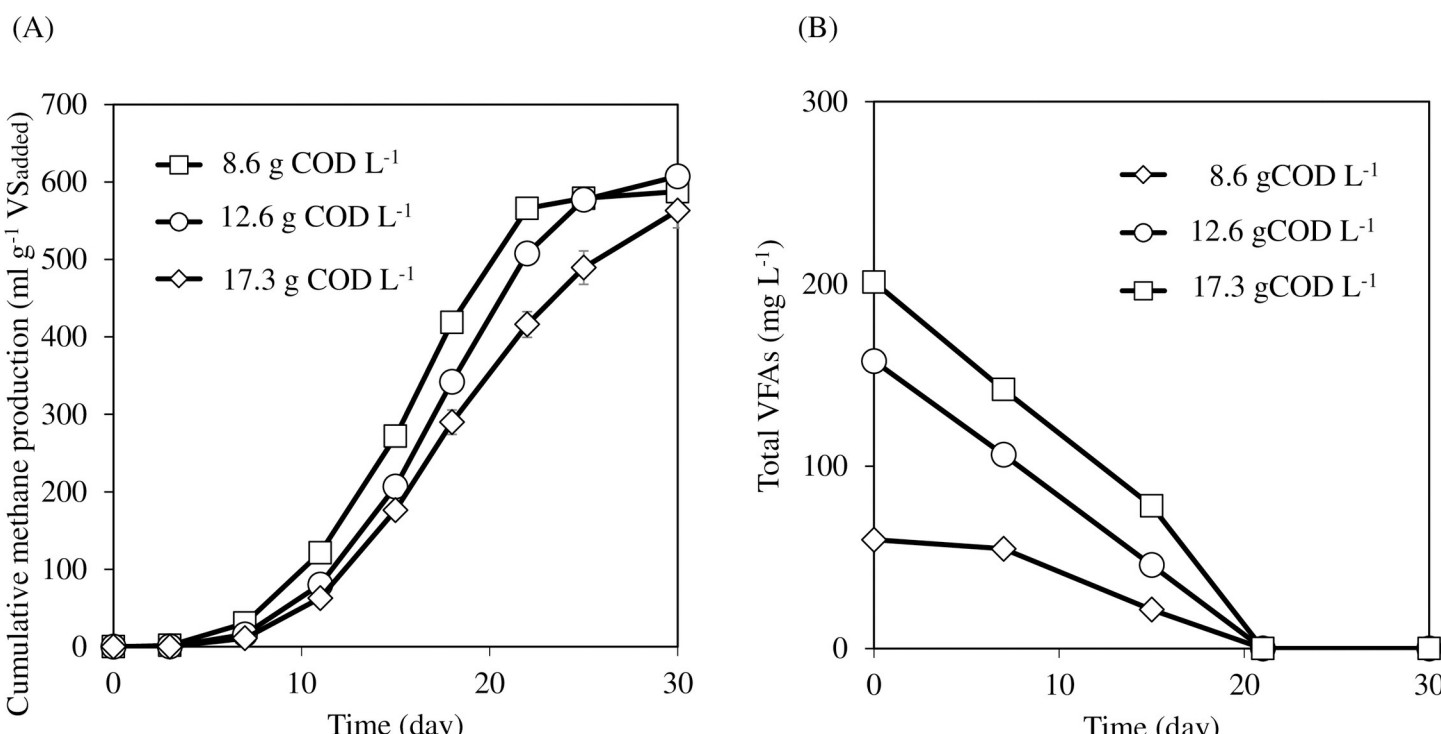

**Fig 4.** Cumulative methane production (A) and VFA concentration (B) at the different initial scum loading concentrations. Error bars represent the standard deviation of the mean (n = 3).

LCFA concentrations in 12.6 and 17.3 g COD L$^{-1}$ loaded vials were much higher than that in Sludge II (Experiment I), in which VFA accumulation and lower methane productivity was observed. Even though high concentrations of LCFAs were detected in scum high-loaded vials, inhibitory effects on scum degradation and methane productivity were not observed in comparison with 8.6 g COD L$^{-1}$ loaded vials. VS decomposition rates were 73% ± 1%, 74% ± 1%, and 71% ± 2% in 8.6, 12.6, and 17.3 g COD L$^{-1}$ loaded vials, respectively (S3A Fig) (TukeyHSD, p >0.05). Dissolved COD in each vial gradually decreased. Finally, the dissolved COD concentrations in 8.6, 12.6, and 17.3 g COD L$^{-1}$ loaded vials were 68.3 ± 5.9, 94.7 ± 2.0, and 95.3 ± 8.0 mg L$^{-1}$, respectively (S3B Fig) (TukeyHSD, p >0.05). These results suggested that the scum digestion process was not inhibited in high-loaded samples. The inhibition caused by LCFAs has been reported as a reversible process [29]. In this study, LCFA-oxidizing bacteria may have degraded the LCFAs adsorbed onto methanogens smoothly. Methanogens and *Syntrophomonas* were quantified to assess the effect of various loading rates on these microbes.

**Quantitative dynamics of methanogens and *Syntrophomonas* by qPCR.** 16S rRNA gene copies for group-specific methanogens were quantified using real-time qPCR (Table 2). Hydrogenotrophic methanogens reduce the partial pressure of hydrogen and enable syntrophic bacteria to proceed with fatty acid oxidation [1]. Methanomicrobiales and Methanobacteriales, which are hydrogenotrophic methanogens, gradually increased in all the samples. At the beginning of the batch experiment, the copy numbers of the 16S rRNA genes belonging to Methanomicrobiales were 7.75 ± 0.10 log copies mL$^{-1}$. On day 8, the copy numbers in 8.6, 12.6, and 17.3 g COD L$^{-1}$ scum loaded vials were 8.63 ± 0.05, 8.67 ± 0.14, and 8.72 ± 0.04 log copies mL$^{-1}$, respectively. On day 30, those in 8.6, 12.6, and 17.3 g COD L$^{-1}$ scum-loaded vials were 8.65 ± 0.06, 9.01 ± 0.05, and 9.06 ± 0.07 log copies mL$^{-1}$, respectively. The 16S rRNA gene concentration belonging to Methanomicrobiales was significantly higher in 12.6 and 17.3 g COD L$^{-1}$ scum-loaded vials than those in 8.6 g COD L$^{-1}$ loaded vials (TukeyHSD, p <0.05). On day 0, the 16S rRNA gene concentration belonging to Methanobacteriales was 5.47 ± 0.01

**Table 2. The transition of 16S rRNA copies for acetoclastic and hydrogenotrophic methanogens (log copies mL$^{-1}$).**

| | 8.6 g COD L$^{-1}$ | 12.6 g COD L$^{-1}$ | 17.3 g COD L$^{-1}$ |
|---|---|---|---|
| *Methanomicrobiales* | | | |
| Day 0 (seed sludge) | 7.75 ± 0.10 | | |
| Day 8 | 8.63 ± 0.05 | 8.67 ± 0.14 | 8.72 ± 0.04 |
| Day 30 | 8.65 ± 0.06[a] | 9.01 ± 0.05[b] | 9.06 ± 0.07[b] |
| *Methanobacteriales* | | | |
| Day 0 (seed sludge) | 5.47 ± 0.01 | | |
| Day 8 | 5.88 ± 0.02[a] | 6.10 ± 0.02[b] | 6.09 ± 0.03[b] |
| Day 30 | 6.71 ± 0.09 | 6.70 ± 0.03 | 6.76 ± 0.05 |
| *Methanosarcinaceae* | | | |
| Day 0 (seed sludge) | 5.64 ± 0.05 | | |
| Day 8 | 6.38 ± 0.07 | 6.12 ± 0.03 | 6.13 ± 0.04 |
| Day 30 | 9.20 ± 0.05 | 9.26 ± 0.03 | 9.37 ± 0.04 |
| *Methanosaetaceae* | | | |
| Day 0 (seed sludge) | 7.89 ± 0.01 | | |
| Day 8 | 7.06 ± 0.02[a] | 7.38 ± 0.05[b] | 7.21 ± 0.04[ab] |
| Day 30 | 9.92 ± 0.11 | 9.95 ± 0.03 | 10.08 ± 0.04 |

All values represent the mean of triplicate reactors ± standard deviation.

Different letters indicate significant differences on the same day (TukeyHSD, p <0.05).

log copies mL$^{-1}$. On day 8, the copy numbers in 8.6, 12.6, and 17.3 g COD L$^{-1}$ scum-loaded vials were 5.88 ± 0.02, 6.10 ± 0.02, and 6.09 ± 0.03 log copies mL$^{-1}$, respectively. The 16S rRNA gene concentration belonging to Methanobacteriales in 12.6 and 17.3 g COD L$^{-1}$ scum-loaded vials was significantly higher than that in 8.6 g COD L$^{-1}$ loaded vials (TukeyHSD, p <0.05). On day 30, the concentrations in 8.6, 12.6, and 17.3 g COD L$^{-1}$ scum-loaded vials were 6.71 ± 0.09, 6.70 ± 0.03, and 6.76 ± 0.05 log copies mL$^{-1}$, respectively. In summary, the copy numbers of the 16S rRNA genes for Methanomicrobiales or Methanobacteriales were significantly higher in scum high-loaded samples on days 8 and 30. It was inferred that hydrogenotrophic methanogenesis was the key process during anaerobic scum degradation.

Methanosarcinaceae and Methanosaetaceae, which are acetoclastic methanogens, showed different quantitative transitions. The copy number of the 16S rRNA genes belonging to Methanosarcinaceae was 5.64 ± 0.05 log copies mL$^{-1}$ on day 0. The copy numbers in 8.6, 12.6, and 17.3 g COD L$^{-1}$ scum-loaded vials were 6.38 ± 0.07, 6.12 ± 0.03, and 6.13 ± 0.04 log copies mL$^{-1}$ on day 8 and 9.20 ± 0.05, 9.26 ± 0.03, and 9.37 ± 0.04 log copies mL$^{-1}$ on day 30, respectively. The 16S rRNA gene copy numbers of Methanosarcinaceae increased gradually in all the samples, and there were no significant differences with respect to scum-loading rates (TukeyHSD, p >0.05). Meanwhile, the copy number of the Methanosaetaceae 16S rRNA gene on day 0 was 7.89 ± 0.01 log copies mL$^{-1}$. On day 8, the copy numbers in 8.6, 12.6, and 17.3 g COD L$^{-1}$ scum-loaded vials were 6.38 ± 0.07, 6.12 ± 0.03, and 6.13 ± 0.04 log copies mL$^{-1}$, respectively. On day 30, those in 8.6, 12.6, and 17.3 g COD L$^{-1}$ scum-loaded vials were 9.92 ± 0.11, 9.95 ± 0.03, and 10.08 ± 0.04, respectively. The 16S rRNA gene copy number of Methanosaetaceae decreased from day 0 to day 8 in all the samples. Methanosaetaceae prefers low acetate concentration [31]. It was simulated that Methanosaetaceae would compete for acetate utilization with Methanosarcinaceae when the acetate concentration was lower than 522.9 mg L$^{-1}$ [32]. Besides, Methanosaeta has been reported to be more sensitive to environmental stress such as ammonia toxicity and overcharging the loading rate than Methanosarcina [31]. Because the VFA concentration detected in this study was kept low (S2A Fig), other stress seemed to inhibit the growth of Methanosaetaceae in the early stage of scum digestion. Through the experiment, the effect of the scum-loading rate on Methanosarcinaceae and Methanosaetaceae was not observed clearly. The high-loaded scum may have restricted the growth of acetoclastic methanogens.

In anaerobic scum digestion, LCFA degradation seemed to be an important process that affects the entire digestion speed and methane conversion rate. 16S rRNA gene copies for *Syntrophomonas* increased in all the samples during the 30-day incubation (Table 3). At the beginning of the batch experiment, the copy number of the 16S rRNA genes for *Syntrophomonas* was 5.56 ± 0.10 log copies mL$^{-1}$. On day 8, the copy numbers in 8.6, 12.6, and 17.3 g COD L$^{-1}$ scum-loaded vials were 6.71 ± 0.06, 6.80 ± 0.03, and 6.59 ± 0.04 log copies mL$^{-1}$, respectively. On day 30, 16S rRNA gene copy number for *Syntrophomonas* in 8.6, 12.6, and 17.3 g COD L$^{-1}$ scum-loaded vials were 7.29 ± 0.04, 7.55 ± 0.03, and 7.77 ± 0.07 log copies mL$^{-1}$, respectively.

**Table 3. The transition of 16S rRNA copies for *Syntrophomonas* (log copies mL$^{-1}$).**

|  | 8.6 g COD L$^{-1}$ | 12.6 g COD L$^{-1}$ | 17.3 g COD L$^{-1}$ |
|---|---|---|---|
| *Syntrophomonas* |  |  |  |
| Day 0 (seed sludge) | 5.56 ± 0.10 |  |  |
| Day 8 | 6.71 ± 0.06 | 6.80 ± 0.03 | 6.59 ± 0.04 |
| Day 30 | 7.29 ± 0.04[a] | 7.55 ± 0.03[ab] | 7.77 ± 0.07[b] |

All values represent the mean of triplicate reactors ± standard deviation.

Different letters indicate significant differences on the same day (TukeyHSD, p <0.05).

On day 30, 16S rRNA gene copies for *Syntrophomonas* observed in 17.3 g COD L$^{-1}$ loaded vials were 3.15-times higher than those in 8.6 g COD L$^{-1}$ loaded vials (TukeyHSD, p <0.05). Similar to hydrogenotrophic methanogens, *Syntrophomonas* abundance increased as the scum-loading rate increased. These results suggest that syntrophic fatty acid degradation by *Syntrophomonas* and hydrogenotrophic methanogens was a key process in scum degradation. It was reported that the abundance of *Syntrophomonas* was correlated with the specific mineralization rates of LCFA [10, 16]. Even in anaerobic scum digestion, the 16S rRNA concentration of *Syntrophomonas* may be used as an indicator of the scum-degrading potential. The transition of *Syntrophomonas* 16S rRNA concentrations was analyzed expecting that LCFA degradation was one of the key processes to determine the efficiency of methane production of scum. The results of Experiment 1 suggested that other complex substances such as protein- or polysaccharide-degrading kinetics, also affected the methane conversion rate. Next, the differences in bacterial communities were analyzed with respect to various scum-loading rates.

**The effect of scum-loading concentration on bacterial community.** The V3-V4 region of the 16S rRNA gene was amplified and sequenced. A total of 48,529–52,393 non-chimeric reads (average 49,957) were obtained from the raw MiSeq data. PCoA suggested that the bacterial communities in 12.6 and 17.3 g COD L$^{-1}$ loaded vials were relatively similar (S4 Fig). Fig 5A shows the top 10 abundant phyla belonging to bacteria. In 8.6 g COD L$^{-1}$ loaded vials, Firmicutes (22.1%) were most abundant and followed by Proteobacteria (18.9%), Bacteroidetes (15.2%), and Cloacimonetes (9.7%). In 12.6 g COD L$^{-1}$ loaded vials, Firmicutes (31.4%) were most abundant, followed by Bacteroidetes (19.3%), Cloacimonetes (9.8%), Proteobacteria (9.5%), and Synergistetes (8.9%). In 17.3 g COD L$^{-1}$ loaded vials, Firmicutes (30.2%) were most abundant, followed by Bacteroidetes (20.7%), Proteobacteria (11.7%), Synergistetes (10.6%), and Cloacimonetes (6.4%). Firmicutes, Bacteroidetes, and Synergistetes were more predominant in scum high-loaded vials. These phyla contain hydrolytic and fermentative bacteria [1] and have been reported to play an important role in the co-digestion of fats, oil, and grease [1,

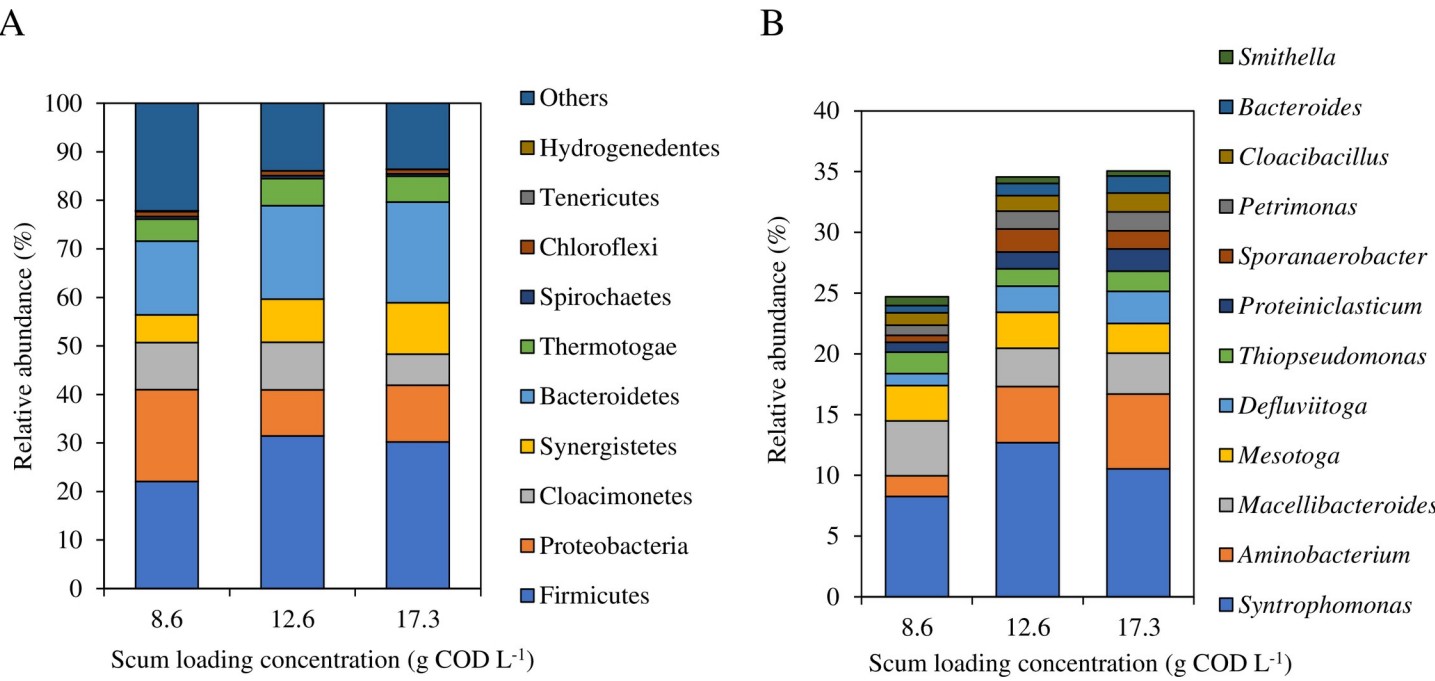

**Fig 5. Relative abundance of bacterial groups based on 16S rRNA genes amplicon sequencing on day 8.** The 10 most abundant phyla (A) and the 12 dominant genera (B) are shown.

33]. Firmicutes include several syntrophic bacteria which degrade various substrates and produce VFAs [34]. Bacteroidetes include various microbial genera that secrete various hydrolytic enzymes and disintegrate complex organic matter [35]. Synergistetes contain only Synergistaceae, which can ferment glucose and organic acids [36]. It was inferred that these phyla contributed to the hydrolysis of complex biomass and degrade the hydrolysate to VFAs.

The top 12 most abundant genera are shown in Fig 5B. In 8.6 g COD L$^{-1}$ loaded vials, *Syntrophomonas* was the most abundant genus (8.3%) and followed by *Macellibacteroides* (4.5%), *Mesotoga* (2.9%), and *Aminobacterium* (1.7%). *Macelibacteroides* use the hydrolysate of polysaccharides as a substrate to produce acetic acid [37]. *Mesotoga* is reported to degrade a wide range of sugars [38]. In 12.6 g COD L$^{-1}$ and 17.3 g COD L$^{-1}$ loaded vials, *Syntrophomonas* was the most abundant genus (12.7% and 10.5%) and followed by *Aminobacterium* (4.6% and 6.2%), *Macellibacteroides* (3.1% and 3.4%), and *Mesotoga* (3.0% and 2.4%). Among the top 12 abundant genera in 12.6 g COD L$^{-1}$ and 17.3 g COD L$^{-1}$ loaded vials, the relative abundance of *Aminobacterium*, *Defluviitoga*, and *Sporanaerobacter* was more than twice as high as that in 8.6 g COD L$^{-1}$ loaded vials (Fig 6). *Aminobacterium* utilizes several amino acids to produce

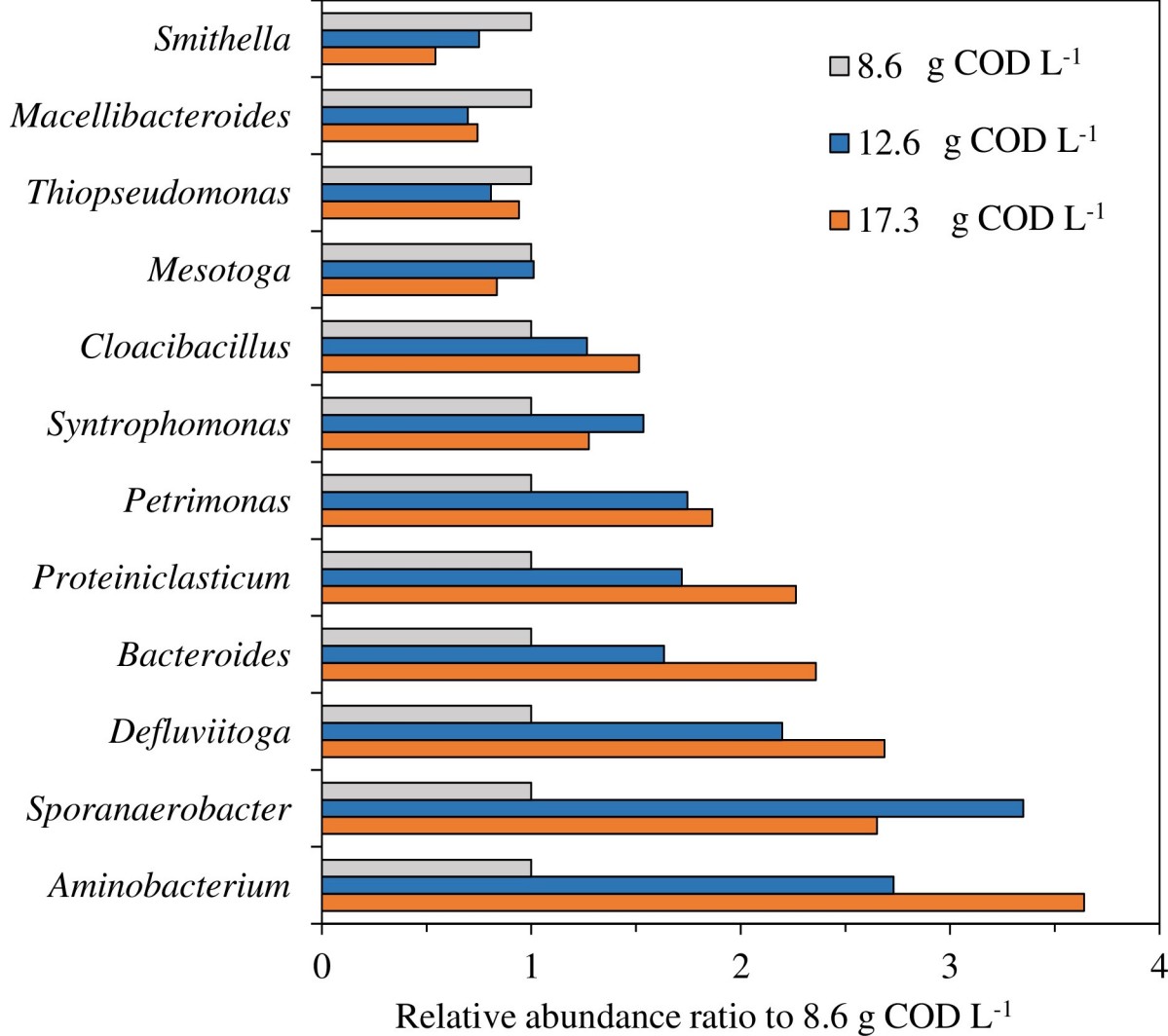

**Fig 6. Comparison of the relative abundance among the 12 dominant genera between 8.6 g COD L$^{-1}$ and 12.6 or 17.3 g COD L$^{-1}$ scum-loaded vials.**

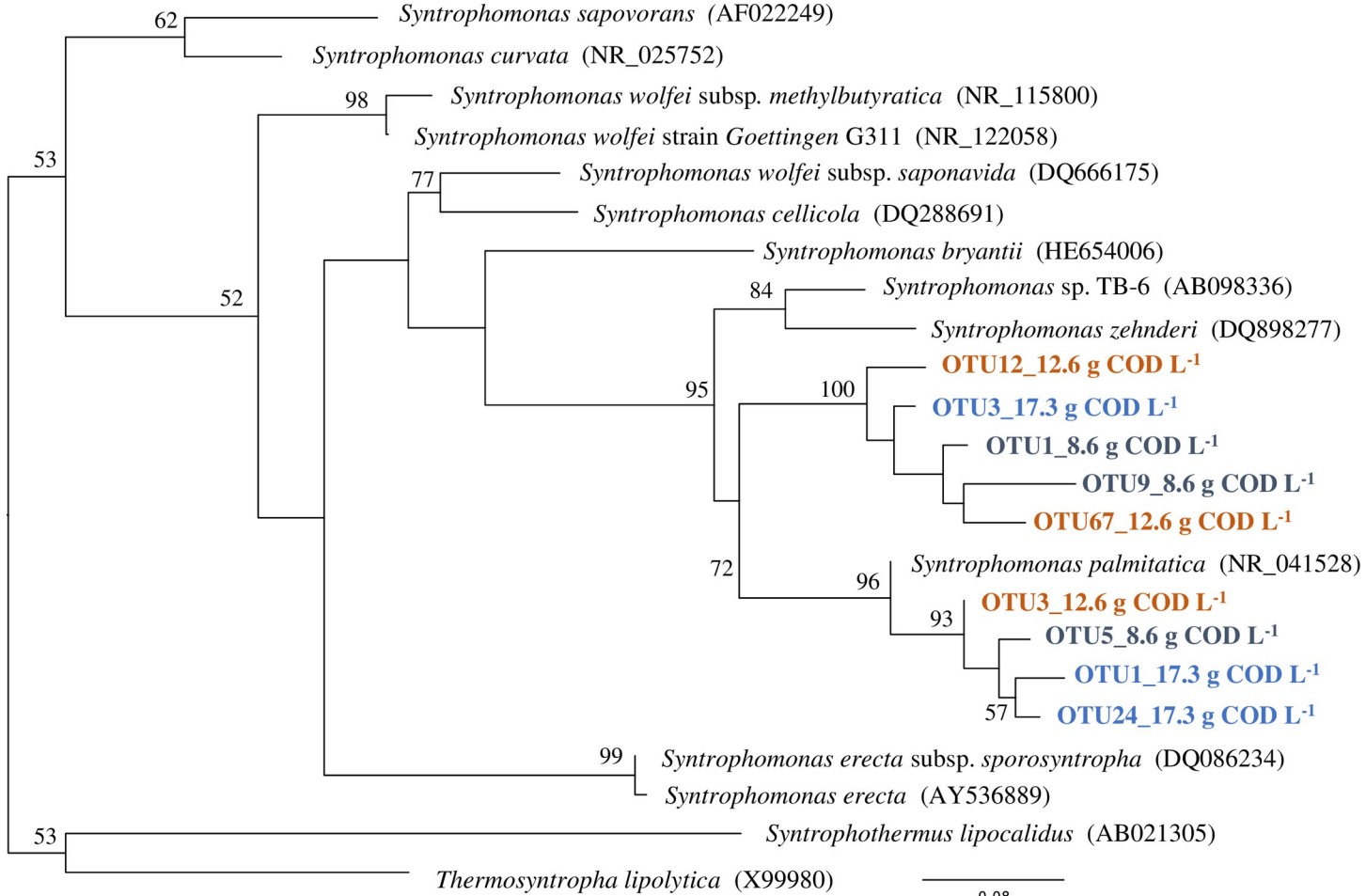

**Fig 7. Phylogenetic tree based on 16S rRNA gene representing the top 3 abundant sequences in the taxon classified as *Syntrophomonas*.** The significance of each branch is indicated at the nodes by bootstrap values (%) based on 100 replications. Only values greater than 50% are shown. The numbers following the scum loading concentration indicate the size of OTUs.

VFAs and ammonia [39]. *Defluviitoga*, which has often been detected in thermophilic biogas plants as the key hydrolytic bacterium, can utilize a large diversity of monosaccharides, disaccharides, and polysaccharides including cellulose and xylan [40]. *Sporanaerobacter* is reported to play a key role in the hydrolysis of proteins and polysaccharides [41]. Other hydrolytic genera also showed higher relative abundance in scum high-loaded samples. *Proteiniclasticum*, which hydrolyze proteins [42], showed a 1.7- and 2.3-times higher abundance in 12.6 and 17.3 g COD L$^{-1}$ loaded vials, respectively. A previous report indicated that *Proteiniclasticum* became dominant by acclimation to fats, oil, and grease [34]. Bacteroides, which play a key role in the initial hydrolysis of protein, fat, cellulose, and other polysaccharides [43], showed a 1.6- and 2.4-times higher abundance in 12.6 and 17.3 g COD L$^{-1}$ loaded vials, respectively. These results suggested that not only LCFA degradation but also the hydrolysis of complex organic matter such as proteins and polysaccharides was an important process in anaerobic scum digestion. Focusing on the abundance of *Syntrophomonas* may not be enough to promote methane gasification of scum. It is also important to increase the abundance of genera shown in this study.

The predominance of *Syntrophomonas* indicated the importance of syntrophic LCFA oxidation. It has been reported that five *Syntrophomonas* species can degrade LCFAs when they

**Table 4. Top 3 abundant taxa in each sample.**

| OTU ID | Closest species of 16S rRNA gene | Oxidizable fatty acids length[a] | Accession no. | Identity | % of sequences[b] |
|---|---|---|---|---|---|
| OTU5 | *Syntrophomonas* | C4-18 | AB274040 | 462/466 | 26.3 |
| 8.6 g COD L$^{-1}$ | *palmitatica* | | | (99%) | |
| OTU1 | *Syntrophomonas* | C4-18 | AB274040 | 443/466 | 15.8 |
| 8.6 g COD L$^{-1}$ | *palmitatica* | | | (95%) | |
| OTU9 | *Syntrophomonas* | C4-18 | AB274040 | 439/466 | 5.6 |
| 8.6 g COD L$^{-1}$ | *palmitatica* | | | (94%) | |
| OTU3 | *Syntrophomonas* | C4-18 | AB274040 | 462/466 | 30.9 |
| 12.6 g COD L$^{-1}$ | *palmitatica* | | | (99%) | |
| OTU12 | *Syntrophomonas* | C4-18 | AB274040 | 442/466 | 7.1 |
| 12.6 g COD L$^{-1}$ | *palmitatica* | | | (95%) | |
| OTU67 | *Syntrophomonas* | C4-18 | AB274040 | 439/466 | 7.0 |
| 12.6 g COD L$^{-1}$ | *palmitatica* | | | (94%) | |
| OTU1 | *Syntrophomonas* | C4-18 | AB274040 | 464/466 | 38.9 |
| 17.3 g COD L$^{-1}$ | *palmitatica* | | | (99%) | |
| OTU3 | *Syntrophomonas* | C4-18 | AB274040 | 446/466 | 12.3 |
| 17.3 g COD L$^{-1}$ | *palmitatica* | | | (96%) | |
| OTU24 | *Syntrophomonas* | C4-18 | AB274040 | 460/466 | 6.1 |
| 17.3 g COD L$^{-1}$ | *palmitatica* | | | (99%) | |

The best hits of the cultured microorganisms are given.

[a] The number of oxidizable carbon chain lengths of straight-chain saturated fatty acids when co-cultured with hydrogenotrophic methanogens.

[b] % of sequences assigned to the *Syntrophomonas* genus.

are co-cultured with hydrogenotrophic methanogens [44]. Ziels et al (2017) showed that the 16S rRNA gene concentration of several OTUs classified as *Syntrophomonas* indicated a closer relationship with oleate degradation kinetics than total *Syntrophomonas* [45]. Species-level analysis would enable us to evaluate the capacity of LCFA degradation more precisely. The sequences assigned to the genus *Syntrophomonas* were aligned into OTUs, and a phylogenetic tree was constructed. Fig 7 shows the top three dominant OTUs of each sample. The dominant OTUs in 8.6, 12.6, and 17.3 g COD L$^{-1}$ scum-loaded vials were closest to *S. palmitatica*, which can oxidize straight-chain saturated fatty acids with carbon chain lengths of C4-C18 [46]. Including uncultured bacteria, OTU1 and OTU9 in 8.6 g COD L$^{-1}$, OTU12 and OTU67 in 12.6 g COD L$^{-1}$, and OTU3 in 17.3 g COD L$^{-1}$ were closest to the clone PM63 (DQ459214) (Identity >97%), which was detected predominantly by DGGE analysis in the oleate and palmitate enrichment cultures [47]. As shown in Table 4, it is noteworthy that the sum of the top 3 abundant OTUs, which were the most closely related to *S. palmitatica*, accounted for 67% of the reads in 17.3 g COD L$^{-1}$ loaded vials. These OTUs possibly played an important role in LCFA degradation. Our results showed that the higher loading rates of scum induced a higher abundance of syntrophically LCFA-oxidizing and complex substrate-hydrolyzing microorganisms.

## Conclusions

This study aimed to explore the microbial community that played a key role in anaerobic scum digestion. The pre-incubated sludge with scum (Sludge I) showed a 73% ± 3% methane conversion rate, which was 1.3-times higher than that of Sludge II, and temporal VFA accumulation was not observed. The relatively high scum degradation potential of this sludge was

confirmed. It was suggested that the degradation efficiency of not only LCFAs but also of other complex substrates such as proteins and polysaccharides affected the methane yield from scum. As the scum-loading rate increased, the larger 16S rRNA copy numbers of *Syntrophomonas* and hydrogenotrophic methanogens were detected. *Aminobacterium*, *Defluviitoga*, and *Sporanaerobacter*, which degrade protein- or polysaccharide-related substrates, also became more abundant as the scum loading amount increased. Phylogenetic analysis of the *Syntrophomonas* genus revealed that the predominant OTUs were close to *S. palmitatica*, which can degrade LCFAs. A higher abundance of *S. palmitatica*-related species in 17.3 g COD L$^{-1}$ scum-loaded vials was observed. These likely played an important role in LCFA degradation. Overall, our results indicate that scum degradation is a complex process in which a variety of genera are involved. To promote scum degradation, it should not only be focused on *Syntrophomonas* but also on the hydrolytic genera suggested in this study.

## Supporting information

**S1 Fig.** VS decomposition rate (A) and transition of dissolved COD concentration (B). Error bars represent the standard deviation of the mean (n = 3).
(TIF)

**S2 Fig. LCFA concentration at the different initial scum loading concentrations.** Error bars represent the standard deviation of the mean (n = 3).
(TIF)

**S3 Fig.** VS decomposition rate (A) and transition of dissolved COD concentration (B) at the different initial scum loading concentrations. Error bars represent the standard deviation of the mean (n = 3).
(TIF)

**S4 Fig. Principal coordinate analysis of 16S rRNA gene amplicon sequences.**
(TIF)

**S1 Table. Characteristics of qPCR primers and TaqMan probes used in this study.** [a]
F = forward primer; P = TaqMan probe; R = reverse primer. [b] $T_m$ of each oligo was cited from its reference article.
(TIF)

## Acknowledgments

We thank Mr. Yoshimi Yokoyama of the Field Science Center, Graduate School of Agricultural Science, Tohoku University, for his technical support in the chemical analyses.

## Author Contributions

**Conceptualization:** Chika Tada.

**Data curation:** Riku Sakurai.

**Formal analysis:** Riku Sakurai.

**Funding acquisition:** Chika Tada.

**Investigation:** Riku Sakurai.

**Methodology:** Riku Sakurai, Shuhei Takizawa, Yasuhiro Fukuda.

**Project administration:** Chika Tada.

**Resources:** Riku Sakurai.

**Validation:** Riku Sakurai.

**Visualization:** Riku Sakurai.

**Writing – original draft:** Riku Sakurai.

**Writing – review & editing:** Shuhei Takizawa, Yasuhiro Fukuda, Chika Tada.

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
