## [Decision Letter · Decision Letter 0]

18 Jun 2021

PONE-D-21-16045

Exploration of the microbial communities contributing to effective methane production from scum under anaerobic digestion

PLOS ONE

Dear Dr. Sakurai,

Thank you for submitting your manuscript to PLOS ONE. After careful consideration, we feel that it has merit but does not fully meet PLOS ONE’s publication criteria as it currently stands. Therefore, we invite you to submit a revised version of the manuscript that addresses the points raised during the review process.

We look forward to receiving your revised manuscript.

Kind regards,

Leonidas Matsakas

Academic Editor

PLOS ONE

Journal Requirements:

Reviewers' comments:

Reviewer's Responses to Questions

**Comments to the Author**

1. Is the manuscript technically sound, and do the data support the conclusions?

Reviewer #1: Yes

Reviewer #2: Partly

Reviewer #3: Yes

2. Has the statistical analysis been performed appropriately and rigorously? 

Reviewer #1: No

Reviewer #2: No

Reviewer #3: No

3. Have the authors made all data underlying the findings in their manuscript fully available?

Reviewer #1: Yes

Reviewer #2: Yes

Reviewer #3: No

4. Is the manuscript presented in an intelligible fashion and written in standard English?

Reviewer #1: Yes

Reviewer #2: No

Reviewer #3: No

5. Review Comments to the Author

Reviewer #1: PONE-D-21-16045

The authors documented the microbial community responsible for the scum degradation in anaerobic sludge. This strategy could significantly degrade the scum with a methane production efficiency of 73%. This also aided the degradation of proteins and polysaccharides along with long-chain fatty acids. Changes in microbial community structure clearly showed the potential of adopted sludge and applied strategy for scum degradation. The study may be considered after addressing the comments during revision.

Comments:

• Did the study compared with control operation-Consumption of the substrate was confirmed by the cessation of biogas generation. Cessation of biogas cannot be considered the only parameter to judge the influence of scum in the digester.

• Line 233-Considering the low methane production in Sludge II, it was inferred that acetate conversion to methane was inhibited till Day 15

• Did Degradation of oleic acid and palmitic acid observed in this study-if degraded then what is the end product in the digester.

• What is more recommended – i) treatment of scum in physical or chemical process with enhanced methane (>50%) efficiency or ii) biological treatment of scum with lower methane conversion rate of 26 % as shown in this study.

• What could be the non-dissolved and complex substances in the undigested sludge?

Reviewer #2: The present manuscript deals with the analysis of the microbial communities which are involved during batch anaerobic digestion of rich-in oils and grease, scum. The study comes into the aims and scopes of the journal and presents a detail analysis of the microbial community, however there are some points that should be addressed, before publication.

The novelty of the study is not presented clearly. In addition, some measurements are missing. Moreover, the authors should have performed blank experiments (only inoculum of sludge 1 and sludge 2 without scum) the methane of which should have been subtracted by the methane of the scum and also microbial analysis should have been performed in that experiments so as direct comparison with the microbial identification of scum, should have been conducted.

Minor comments

1) The authors should be careful with the English writing and diction and the whole presentation of their manuscript. There are so many grammatical and syntax errors.

2) The abstract needs revision to be more clear

3) The authors use first plural form in order to describe what they have done. To my opinion, is better to present the methodology and results in third singular

4) Line 64: disposal of scum does not lead to biomass washout. Please revise

5) Lines 66-67, 71-71, line 74, lines 228-229: syntax problems

6) Line 86: grammatical errors

7) Lines 91-94: not well written. not apparent the innovation. The authors should give the research gap that their work fills in.

8) line 108: which are the characteristics of sludge in terms of COD, pH TS and VS? please give a table

9) line 108: Was there any protocol that was followed?

10) which was the reason of developing the sludge 2? Please comment.

11) Line 115: experiments, line 117: better 40 mL

12) which was the protocol followed for the batch methane experiments? Please provide a table with the characteristics of scum and sludge

13) the authors should have performed blank and control experiments.

14) Lines 129-131: not clear

15) Line 216, 218: methane production of sludge

16) Line 248: units?

17) Lines 252-253: repeat

18) Line 253: enzymes

19) Table 1: units? line 299: units?

20) Line 309 :syntax

21) Lines 395-305: syntax errors and repeat

22) Line 405: bacteria , line 411? viales?

Reviewer #3: The present study aimed to investigate the biomethane productivity of scum waste along with changes in microbial communities with respect to various loading rates. Firstly, enrichment of scum-degrading microbes was evaluated with the collected sludge and scum digestibility was assessed by comparing with another collected sludge. The results showed 1.3 times higher methane conversion rate (73%) and faster LCFA degradation in the pre-incubated sludge with scum. Secondly, the scum loading rates' influence on the microbial community was studied, wherein the increased 16S rRNA copy numbers for the syntrophic fatty-acid degrader, Syntrophomonas, and hydrogenotrophic methanogen were observed in scum high-loaded samples. These results indicated profiles of microorganisms that degrade LCFAs in scum anaerobic digestion. The objective of the study is novel, but the methodology representation and results discussion are not appropriate. Recommending for major revision and below suggested comments could be useful to improve the manuscript quality.

• The abstract is too lengthy and contains general discussion, recommending modify the abstract as per the specific objectives of the study along with including the obtained results.

• Maintain a similar font size throughout the manuscript and give the appropriate numbering to the sections.

• The compositional details of scum such as VS, TS, Total COD, Soluble COD, VFA, and LCFA are is not included.

• The inoculum loading rates in Experiment 1 and 2 are not included.

• The methodology is not clear, the authors were chosen different sludges (Sludge I and II), loading rates, and different batch studies. Suggesting to rewrite the methodology by maintaining the flow also includes a schematic representation of work for better understanding to the readers.

• Important results like VS or COD reduction is missing with respect to operation day.

• The LCFA utilization with respect to operation day in figure format could be useful rather than a table.

• In table 1, the details of AS 8.6 and CS 8.6 are not provided, suggesting to expand such abbreviations at the bottom of the table.

• The discussed results of scum loading concentration influence on methane productivity is too short and have less discussion, suggesting to elaborate the discussion by covering the influence of various loading rates along with the LCFA utilization with respect to operation day.

• Overall, manuscript quality is very poor and needs significant changes with abstract, methodology and results discussion specifically with VS, LCFA utilization. Some of the supplementary figures could be included in the main content with an appropriate discussion.

6. PLOS authors have the option to publish the peer review history of their article (what does this mean?). If published, this will include your full peer review and any attached files.

Reviewer #1: No

Reviewer #2: No

Reviewer #3: No

---

## [Decision Letter · Decision Letter 1]

8 Sep 2021

Exploration of the microbial communities contributing to effective methane production from scum under anaerobic digestion

PONE-D-21-16045R1

Dear Dr. Sakurai,

We’re pleased to inform you that your manuscript has been judged scientifically suitable for publication and will be formally accepted for publication once it meets all outstanding technical requirements.

Kind regards,

Leonidas Matsakas

Academic Editor

PLOS ONE

Additional Editor Comments (optional):

Reviewers' comments:

Reviewer's Responses to Questions

**Comments to the Author**

1. If the authors have adequately addressed your comments raised in a previous round of review and you feel that this manuscript is now acceptable for publication, you may indicate that here to bypass the “Comments to the Author” section, enter your conflict of interest statement in the “Confidential to Editor” section, and submit your "Accept" recommendation.

Reviewer #1: (No Response)

Reviewer #2: All comments have been addressed

2. Is the manuscript technically sound, and do the data support the conclusions?

Reviewer #1: (No Response)

Reviewer #2: Yes

3. Has the statistical analysis been performed appropriately and rigorously? 

Reviewer #1: (No Response)

Reviewer #2: (No Response)

4. Have the authors made all data underlying the findings in their manuscript fully available?

Reviewer #1: (No Response)

Reviewer #2: (No Response)

5. Is the manuscript presented in an intelligible fashion and written in standard English?

Reviewer #1: (No Response)

Reviewer #2: No

6. Review Comments to the Author

Reviewer #1: (No Response)

Reviewer #2: (No Response)

7. PLOS authors have the option to publish the peer review history of their article (what does this mean?). If published, this will include your full peer review and any attached files.

Reviewer #1: No

Reviewer #2: No

---

## [Editor Report · Acceptance letter]

21 Sep 2021

PONE-D-21-16045R1 

Exploration of microbial communities contributing to effective methane production from scum under anaerobic digestion 

Dear Dr. Tada:

I'm pleased to inform you that your manuscript has been deemed suitable for publication in PLOS ONE. Congratulations! Your manuscript is now with our production department. 

Kind regards, 

on behalf of

Dr. Leonidas Matsakas 

Academic Editor

PLOS ONE